# The role and targeting potential analysis of angiogenesis-related target THY1 in DSS-induced acute colitis in mice

Pengliang Zhang[1], Xianmin Liu[1], Shuang Chen[2], Yingjian Zhang[1]*

1 Division of Gastroenterology, The First Affiliated Hospital of Henan University of Science & Technology, Luoyang, China, 2 Medical Plastic Surgery Center, The First Affiliated Hospital of Henan University of Science & Technology, Luoyang, China

* zhangyingjian@haust.edu.cn

## Abstract

### Background

Immune-driven inflammatory angiogenesis plays a crucial role in the pathogenesis of inflammatory bowel disease (IBD). However, the mechanism of chronic inflammation mediated by angiogenesis still remains unclear. This study aimed to investigate the crucial role and specific mechanism of the angiogenesis-related target THY1 in the development of acute colitis induced by dextran sulfate sodium (DSS) in mice.

### Methods

Lentivirus-based systems were utilized to achieve both knockdown and overexpression of THY1 to explore the functional roles of THY1 in IBD development based on DSS-induced colitis mice model and co-culture system of intestinal epithelial cells and macrophages.

### Results

THY1 significantly promotes DSS-induced colitis in the experimental mouse model. Silencing of the THY1 significantly reversed the inflammatory response, oxidative stress level, and angiogenic activity in DSS-induced colitis, whereas overexpression of THY1 further exacerbating the reactions related to inflammation, oxidative stress, and angiogenesis. Mechanism research showed that THY1 promotes the pathological process of DSS-induced colitis through inhibiting M2 macrophage polarization. In addition, THY1 promotes apoptosis through the Bcl-2/Bax/Cleaved caspase-3 pathway and promotes angiogenesis via upregulation of HIF-1α and VEGF expression in DSS-induced colitis.

**Data availability statement:** All relevant data are within the manuscript and its Supporting information files.

**Funding:** The author(s) received no specific funding for this work.

**Competing interests:** The authors have declared that no competing interests exist.

## Conclusions

THY1 promotes the occurrence and development of acute colitis induced by DSS in mice by regulating macrophage polarization, intestinal epithelial cell apoptosis, and inflammatory angiogenesis, providing a new perspective for the study of the pathogenesis of IBD. Furthermore, THY1 is expected to become a potential target molecule for the treatment of IBD.

## Introduction

Inflammatory Bowel Disease (IBD), including Crohn's Disease (CD) and Ulcerative Colitis (UC), is a group of diseases characterized by chronic and recurrent gastrointestinal inflammation [1–3]. Its pathogenesis is complex, involving the interaction of multiple factors such as genetic susceptibility, immune dysregulation, gut microbiota disorder, and environmental factors [4,5]. Although the application of biological agents and small-molecule targeted drugs has significantly improved the clinical prognosis of IBD patients in recent years, some patients still have poor treatment responses or develop complications [6]. Therefore, the exploration of novel treatment strategies has become an issue of particular urgency.

Increasing research evidence suggests that immune-driven inflammatory angiogenesis plays a crucial role in the occurrence and development of IBD [7–9]. Under normal physiological conditions, the intestinal vascular network maintains tissue oxygen supply and nutrient exchange through precise regulation, ensuring the stable operation of intestinal functions [10,11]. During the mucosal healing process in the initial stage of IBD, angiogenesis provides indispensable support for tissue repair. It not only promotes the delivery of nutrients but also accelerates the clearance of cellular metabolic waste [12]. However, the inflamed mucosa releases a large number of inflammatory factors, which often lead to tissue damage and further trigger pathological angiogenesis [13]. In addition, the overexpression of pro-angiogenic factors (such as VEGF and TNF-α) drives abnormal blood vessel proliferation, leading to a series of pathological changes such as increased vascular permeability and blood flow disorders [14]. These changes not only further exacerbate tissue hypoxia and inflammatory cell infiltration but may also promote the fibrotic process, thereby increasing the disease burden. Nevertheless, the specific regulatory mechanism of angiogenesis-mediated chronic intestinal inflammation has not been fully elucidated. Whether targeting angiogenesis can become an effective strategy for treating IBD still requires more systematic and in-depth research to clarify its potential clinical value [15]. Exploration in this field not only helps to uncover the complex pathological mechanisms of IBD but may also provide new ideas and directions for future treatment regimens.

Thymocyte antigen-1 (THY1), also known as CD90, is a cell surface protein (25–37 kDa) that anchored to the cell membrane via a glycosylphosphatidylinositol (GPI) anchor, which plays an important role in physiological and pathological processes such as tumor angiogenesis, tissue repair, and immune regulation [16,17].

For instance, Foygel et al. have successfully identified and validated THY1 as a specific biomarker for neoangiogenesis in pancreatic ductal adenocarcinoma, which can be effectively detected through ultrasound molecular imaging, thereby enhancing the diagnostic accuracy of this malignancy and contributing to improved patient management [18]. In addition, Single-cell transcriptomics studies have revealed that THY1+ fibroblasts play a pivotal role as key mediators in driving joint inflammation in rheumatoid arthritis [19,20]. These findings underscore their critical importance in the development of cell-based therapies aimed at modulating inflammation and mitigating tissue damage. Nevertheless, as of the current state of research, the specific role of THY1 in IBD remains unexplored. Furthermore, its potential regulatory mechanisms, particularly its involvement in the pathogenesis of IBD through the modulation of inflammatory angiogenesis and related pathological processes, remain unclear and warrant further investigation. This study aims to explore the potential role of THY1 in the pathological mechanism of IBD from the perspective of HIF-1α/VEGF meditated angiogenesis.

## Materials and methods

### Animal model construction

Female C57BL/6 mice, aged 8 weeks (weighing 20–22 g), were procured from Vital River Laboratory Animal Technology Co., Ltd. Located in Beijing, China. The mice were adaptively maintained in standard cages (330×210×170 mm; 6 mice/cage) for 1 week under standardized housing conditions, which included a controlled ambient temperature of 23±2°C, a relative humidity of 55±5%, and a 12-hour light/dark cycle. And the mice were provided unrestricted access to food and water throughout the study. Thirty-six mice in this study were randomly divided into 6 groups (six mice per group): the Sham (blank control) group, the IBD model group, the IBD+OE-THY1 group, the IBD+OE-NC group. the IBD+KD-THY1 group, and the IBD+sh-NC group. Group randomization was performed using a random number table. The sample size was calculated based on a power analysis combining preliminary data and literature benchmarks.

On the 4th day of the adaptive housing period, intraperitoneal injections were given to the mice in each THY1 regulation group. Lentiviral vectors for THY1 overexpression or inhibition (100 μL, containing approximately 5×10$^8$ PFU) were administered respectively, and the virus titer was adjusted in advance to ensure the consistency of the experiment. The control groups were injected with an equal volume of normal saline (NS) or the corresponding empty vector of the lentivirus. All surgery was performed under sodium pentobarbital anesthesia, and all efforts were made to minimize suffering. After completing the 1 – week adaptive housing (including virus intervention), the drinking water of the mice was replaced with purified water containing 2.5% (w/v) dextran sulfate sodium (DSS) for 7 consecutive days to induce an inflammatory bowel disease (IBD) model. Subsequently, normal drinking water was restored, and the mice were continuously observed until the 10th day of the experiment. During this period, all the mice were maintained in standard cages under standardized housing conditions and provided unrestricted access to food and water. The body weight changes of all mice were recorded daily (from the 1st day to the 10th day of model establishment) to evaluate the disease progression and the effectiveness of model establishment.

Euthanasia will only be carried out before the planned endpoint when the animals meet the pre-defined humane standards. These standards include (but are not limited to): severe weight loss (more than 20% of the baseline body weight) or stunted growth; irreversible signs of pain or suffering (such as shortness of breath, prolonged immobility, inability to access food/water); unexpected complications directly related to the experimental intervention (such as neurological deficits). In this study, no animals needed to be euthanized prematurely because the health indicators of all subjects remained within the pre-set thresholds throughout the experiment. At the end of the experiment, all the mice were anesthetized by intraperitoneal injection of 1% sodium pentobarbital (50 mg/kg) and then sacrificed by cervical dislocation, which took about 10 seconds. Peripheral blood and colon tissues were collected for subsequent analysis. Since the experiment was designed as a terminal experiment, no animals were retained after the study. All experimental protocols involving animals were reviewed and approved by the Committee for Animal Care and Use of The First Affiliated Hospital of Henan

University of Science & Technology (Approval No. D-2025-B034) and conducted in accordance with the National Laboratory Animal Care and Maintenance Guide.

## Histological analysis

H&E staining: Colon tissues were continuously fixed in 40% formaldehyde for 24 hours. They were dehydrated with ethanol, cleared with xylene, embedded in paraffin, and sectioned using a microtome. Paraffin sections of colon tissues were continuously stained with hematoxylin-eosin (H&E), and the tissue damage was observed under an optical microscope.

TUNEL staining: Tissue sections were first treated with xylene to complete the dewaxing step, and then further dewaxed with gradient alcohol to ensure the full preparation of tissue samples. Next, antigen retrieval is performed using citrate buffer to restore the antigen activity in the tissue. To reduce non – specific binding, the sections are blocked with 5% skimmed milk powder. On this basis, the TdT enzyme reaction solution and the streptavidin – fluorescein isothiocyanate (Streptavidin – FITC) labeled working solution are successively applied to the tissue sections, and the TUNEL staining operation is completed according to the standard procedure.

## Cell culture and establishment of the colitis cell model

Normal human intestinal epithelial cell line NCM460 were acquired from American Type Culture Collection (ATCC, Manassas, VA, USA) and cultured in DMEM medium (Invitrogen, Carlsbad, CA, USA) supplemented with 10% fetal bovine serum (Gibco, Waltham, MA, USA) and 100 units of penicillin/streptomycin, and then placed in a constant-temperature incubator at 37°C with 5% $CO_2$ for growth. NCM460 cells were gradient intervention with different concentrations of DSS (e.g., 0, 0.1, 0.2, 0.5, 1, 2 μg/mL) for 12 hours to establish an IBD cell model. The cells and its supernatant were collected and stored at −80 °C for subsequent ELISA, RT-qPCR and Western blotting (WB) assay. The expression levels of inflammatory factors including IL-1β, IL-6, TNF-α, and TGF-β in cells and its supernatant were detected by ELISA to evaluate the disease progression and the effectiveness of model establishment, referring to the published methods [21]. After verifying its inflammation-inducing efficiency on NCM460 cells, a DSS working concentration of 1 μg/ml was selected for the subsequent cell experiments (S1 Fig).

## Cell transfection

The shRNA targeting THY1 (sh-THY1) and the THY1-overexpressing plasmid (pcDNA3.1-THY1), along with their corresponding control constructs, were obtained from GenePharma (Shanghai, China). NCM460 cells were transfected with these constructs using Lipofectamine 2000 (Invitrogen) for a duration of 48 hours, following the manufacturer's recommended protocol. The efficiency of both knockdown and overexpression was subsequently evaluated through qRT-PCR and Western blot analyses.

## Establishment of co-culture system

In this study, a co-culture system of intestinal epithelial cells and macrophages was established using the Transwell chamber system. The specific operations are as follows: untreated NCM460 cells or successfully transfected NCM460 cells were seeded in the lower layer of the Transwell chamber at a density of $1 \times 10^5$ cells/well, while $0.5 \times 10^5$ THP-1 cells that had been induced and differentiated by Phorbol 12-myristate 13-acetate (PMA, 100 ng/ml) for 24 hours were added to the upper layer. Subsequently, DMEM complete medium was added to the system, and DSS with a final concentration of 1 μg/ml was added for 12 hours of culture. After the intervention, the cells and its supernatant from the upper and lower layers of the chamber were collected respectively, and samples were stored at −80 °C for subsequent ELISA, RT-qPCR, Western blotting (WB) and Flow cytometry assays. The co-culture system was divided into 6 groups: Blank group (Upper layer: empty; Lower layer: normal NCM460 cells), Control group (Upper layer: THP-1 cells; Lower layer: normal NCM460

cells), sh-THY1 group (Upper layer: THP-1 cells; Lower layer: sh-THY1 treated NCM460 cells), sh-NC group (Upper layer: THP-1 cells; Lower layer: sh-NC treated NCM460 cells), OE-THY1 group (Upper layer: THP-1 cells; Lower layer: OE-THY1 treated NCM460 cells), and OE-NC group (Upper layer: THP-1 cells; Lower layer: OE-NC treated NCM460 cells).

## Enzyme-linked immunosorbent assay (ELISA)

Pro-inflammatory cytokines (IL-1β, TNF-α, and TGF-β), oxidative stress-related molecules (ROS, MDA and GSH) and angiogenesis-related stimulatory factors (HIF-1 and VEGF) were quantified utilizing the respective ELISA kits (mlbio, Shanghai, China) following the manufacturer's instruction. In short, the samples including mouse colon tissue, mouse serum, cells and its supernatant samples were treated with the reaction solution and termination solution. The absorbance was assessed at a wavelength of 450 nm utilizing a microplate reader (Molecular Devices).

## Quantitative real-time PCR

Total RNA was isolated from cells using TRIzol reagent (Invitrogen). The extracted RNA was reverse transcribed into cDNA using PrimeScript RT Reagent kit (Takara, Dalian, China). Quantitative real-time PCR was conducted with SYBR Premix Ex Taq kit (Takara) on the ABI 7500 real-time PCR system (Applied Biosystems, Foster City, CA, USA) with specific primers according to the manufacturer's instructions. Gene expression was calculated utilizing $2^{-\Delta\Delta Ct}$ method as normalized to GAPDH. Primers used in this study were synthesized by Shanghai Sangon Biotechnology Co., Ltd. (Shanghai, China) and primer sequences are shown in Table 1.

## Western blot

Following the application of the respective treatments, cellular proteins were extracted from each sample by the Radio-Immunoprecipitation Assay (RIPA) lysis buffer (Beyotime Biotechnology, Shanghai, China). Protein concentrations were determined using the BCA (bicinchoninic acid) protein assay kit (Beyotime). Subsequently, the proteins were fractionated on SDS-PAGE gels containing either 10% or 12% acrylamide and then electrophoretically transferred onto PVDF membranes. The membranes were blocked with 5% BSA and subsequently probed with immunoblotting techniques. Primary antibodies like anti-THY1 (14-0909-82, 0.5 mg/mL; Thermo Fisher), anti-Bcl-2 (12789–1-AP, 1:2000; Proteintech), anti-Bax (50599–2-Ig, 1:5000; Proteintech), anti-cleaved caspase-3 (abs132005, 1:500; Absin), anti-HIF-1α (BF8002, 1:1000;

Table 1.  The primer sequences for relative mRNA used in this study.

| Primer name | Sequence (5'-3') |
| --- | --- |
| THY1-human-F | ATCGCTCTCCTGCTAACAGTC |
| THY1-human-R | CTCGTACTGGATGGGTGAACT |
| β-actin-human-F | ACGTGGACATCCGCAAAG |
| β-actin-human-R | TGGAAGGTGGACAGCGAGGC |
| THY1-mouse-F | GCTCTCAGTCTTGCAGGTGTC |
| THY1-mouse-R | CAGGCGAAGGTTTTGGTTCA |
| VEGF-mouse-F | GCTACTGCCGTCCGATTGAGA |
| VEGF-mouse-R | GCTGGCTTTGGTGAGGTTTGAT |
| HIF-1α-mouse-F | GATGACGGCGACATGGTTTAC |
| HIF-1α-mouse-R | CTCACTGGGCCATTTCTGTGT |
| β-actin-mouse-F | GTGACGTTGACATCCGTAAAGA |
| β-actin-mouse-R | GCCGGACTCATCGTACTCC |

Affinity) and anti-VEGF (ab46154, 1:1000; Abcam) were used at the indicated dilutions and incubated at 4°C. After the membrane was washed three times, the secondary antibody goat Anti-Human IgG Fc (HRP) (ab98624, 1:5000; Abcam), was incubated at room temperature for 2 hours. The membrane was scanned and imaged using the ChemiDoc XRS+ imaging system (Bio-Rad, America) after color development with ECL chemiluminescent reagent (Beyotime). For image quantification, the immunoblots were analyzed using ImageJ software and the protein levels were quantified as the ratio of its gray value to that of β-actin (GB11001, 1:2000; Serivicebio).

### Flow cytometry analysis

Macrophage immunophenotyping: the upper THP-1 cells from co-culture system were collected, washed and resuspended in PBS. Then, 5 μL of CD11b-FITC, CD86-APC (M1 marker) or CD206-APC (M2 marker) were added and incubate at room temperature in the dark for 15 min. Subsequently, the M1 or M2 macrophages were enumerated using a FACS Canto II flow cytometer. The gating strategy adopted a step-by-step approach: First, use the forward scatter (FSC)/side scatter (SSC) parameters to exclude debris and doublet cells, and then select live CD11b+ monocytes/macrophages.

Apoptosis of NCM460 cells in co-culture system was assayed using the Annexin V-FITC apoptosis detection kit provided by BD Biosciences. Cells were harvested, rinsed with PBS, and resuspended in 300 microliters of Annexin V binding buffer to achieve a single-cell suspension. Next, 5 microliters of Annexin-V-fluorescein isothiocyanate (FITC) and 5 microliters of propidium iodide (PI) were concurrently added to the cell suspension and incubated in the dark at ambient temperature for a period of 15 minutes. Subsequently, the apoptotic cells were enumerated using a FACS Canto II flow cytometer.

### In vitro tube formation assay

The 96-well plates were pre-coated with a frozen Matrigel solution (phenol red-free type; BD Biosciences, New Jersey, USA) and incubated at 37°C for at least 30 minutes to ensure full solidification of the Matrigel. Subsequently, human umbilical vein endothelial cells (HUVECs) were collected and suspended in a low-serum medium containing 2% fetal bovine serum (FBS), then seeded into the Matrigel-coated wells at a density of $1 \times 10^5$ cells/ml per well and pre-incubated at 37°C for 30 minutes to allow cell attachment. During this process, the cell culture supernatants previously collected from different experimental groups in the co-culture system were used as conditioned media for HUVECs and placed in the cell incubator for further incubation. Images of the formed tubular structures were acquired using a real-time cell imager (NanoEntek, Seoul, South Korea) during the 4–24 hours of regular cell culture.

### Statistical analyses

Statistical evaluations were performed utilizing the GraphPad Prism 8 software package. Unless indicated otherwise, the data presented are expressed as the mean ± SD, derived from a minimum of three separate experiments. Significance testing of the differences between means was conducted using either two-tailed Student's t-tests or ANOVA, with subsequent application of Tukey's post hoc test, as deemed appropriate for the specific dataset.

## Results

### THY1 promotes dextran sulfate sodium (DSS)-induced colitis in mice

In this study, lentivirus-based systems were utilized to achieve both knockdown and overexpression of THY1 to explore the functional roles of THY1 in IBD development. Following the stable transfection of NCM460 cells with THY1 suppression and overexpression vectors, the successful knockdown and overexpression of THY1 expression levels was both confirmed and visually demonstrated (Fig 1A). In DSS-induced colitis mouse model, the administration of 3.0% DSS induced an increased expression of THY1 compared to the control group, and successful knockdown and overexpression

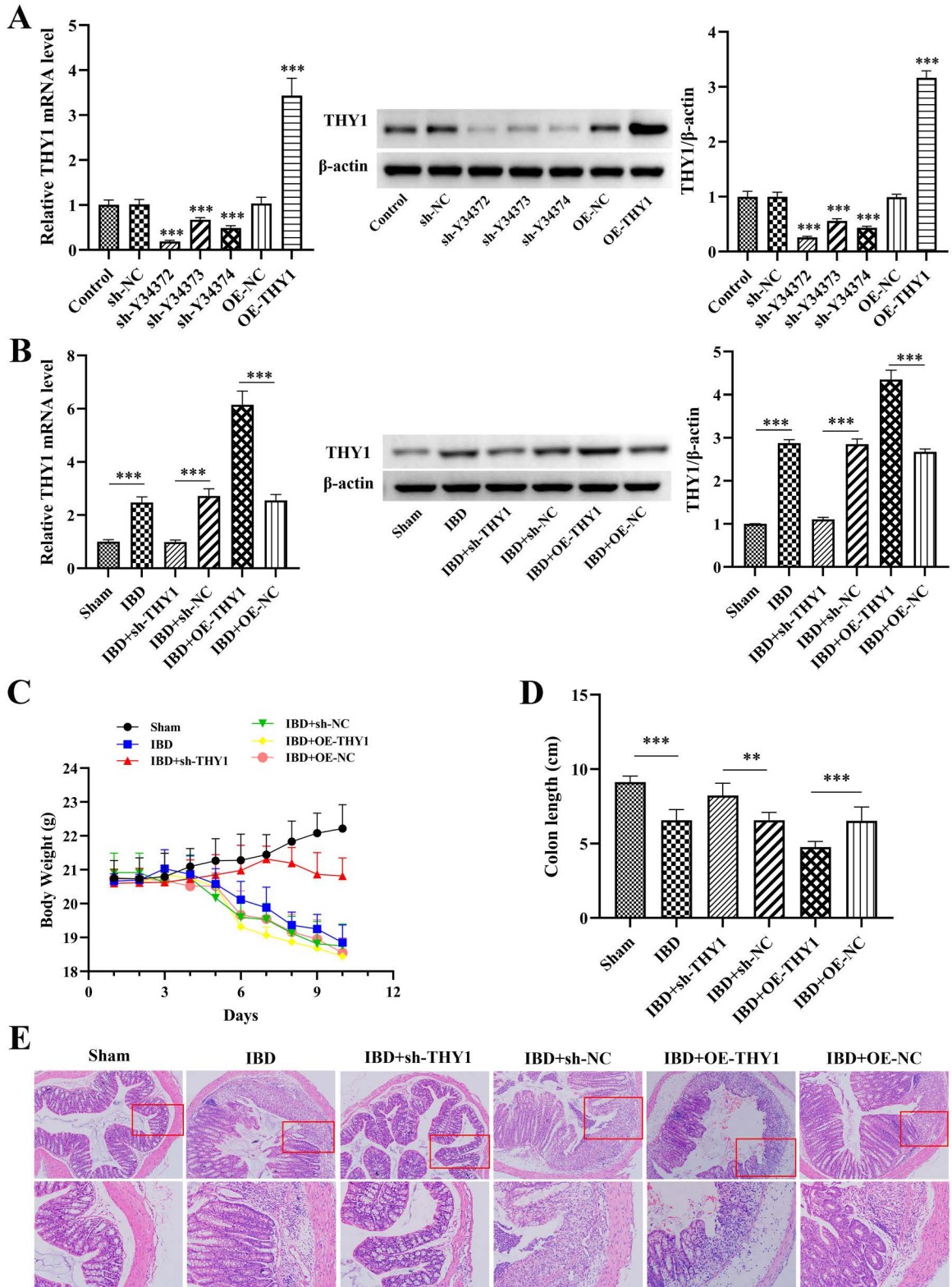

**Fig 1. THY1 promotes dextran sulfate sodium (DSS)-induced colitis in mice.** (A, B) The knockdown and overexpression of THY1 expression levels in NCM460 cells and DSS-induced colitis mouse colon tissue samples were determined by quantitative real-time PCR and western blot analysis,

respectively. (C) Body weight loss and (D) Appearance of colon length of DSS-induced colitis mouse in different treatment groups including the Sham (blank control) group, the IBD model group, the IBD + OE-THY1 group, the IBD + OE-NC group, the IBD + KD-THY1 group, and the IBD + sh-NC group. (E) Representative H&E staining images of DSS-induced colitis mouse colon samples in different treatment groups. The corresponding image below is a high-magnification view of the red square position in the image above. A minimum of three separate experiments were carried out and the data presented are expressed as the mean ± SD. *P < 0.05, **P < 0.01, ***P < 0.001.

of THY1 expression levels were also confirmed and visualized following lentivirus injection via the tail vein with sh-THY1 and OE-THY1 (Fig 1B). As illustrated in Fig 1C and 1D, the administration of 3.0% DSS over a period of 10 days resulted in significant weight loss and a marked reduction in colon length compared to the control group, whereas knockdown of THY1 effectively alleviated these adverse effects while overexpression of THY1 significantly aggravated the symptoms of enteritis. In addition, histological examination of colon sections showed that the samples from mice treated with DSS + sh-THY1 exhibited well-preserved colon structure, no obvious ulcers, and reduced inflammatory cell infiltration in contrast to that observed in DSS and DSS + sh-NC groups, while the samples from mice treated with DSS + OE-THY1 showed further damaged colon structure with obvious ulcers and an increase in inflammatory cell infiltration compared to DSS and DSS + OE-NC groups (Fig 1E). These findings collectively demonstrate that THY1 significantly promotes DSS-induced colitis in the experimental mouse model.

### THY1 promotes the pathological process of DSS-induced colitis and inhibits M2 macrophage polarization

In order to deeply explore the specific mechanism by which THY1 plays a role in DSS-induced colitis, this study conducted multi-level analyses on mouse peripheral blood and colon tissue samples. First, the expression levels of inflammatory factors, oxidative stress-related molecules and angiogenesis-related stimulatory factors were systematically detected to evaluate their dynamic changes during the pathological process. Compared with the control group, the expression of inflammatory factors IL-1β and TNF-α was significantly upregulated in the DSS treatment group, while the expression of TGF-β was significantly downregulated. Meanwhile, the levels of molecules related to oxidative stress, ROS and MDA, were significantly increased, while the level of the antioxidant molecule GSH was significantly decreased. In addition, the expression of key stimulatory factors related to angiogenesis, HIF-1 and VEGF, also showed a significant upward trend (Figs 2A and S2). These results indicate that there are obvious inflammatory responses, enhanced oxidative stress, and increased angiogenic activity in the DSS-induced colitis model. Further studies found that silencing of the THY1 could significantly reverse the inflammatory response, oxidative stress level, and angiogenic activity in DSS-induced colitis. In contrast, overexpression of THY1 showed the opposite effect, further exacerbating the reactions related to inflammation, oxidative stress, and angiogenesis (Figs 2A and S2). These results suggest that THY1 may play an important regulatory role in the pathological process of DSS-induced colitis.

In addition, we successfully established a co-culture system of macrophages and intestinal epithelial cells utilizing Transwell chambers to further validate the regulatory role of THY1 in the pathological progression of DSS-induced colitis. As shown in Fig 2B, co-culturing with macrophages (Control group) significantly upregulated the expression of inflammatory factors (IL-1β and TNF-α), and at the same time significantly inhibited the level of the anti-inflammatory factor (TGF-β) in NCM460 cell homogenates compared with Blank group. In addition, significant changes also occurred in molecules related to oxidative stress: the levels of ROS (reactive oxygen species) and MDA (malondialdehyde) increased significantly, while the content of GSH (glutathione) decreased significantly. Meanwhile, the expression of key factors related to angiogenesis, HIF-1 and VEGF, also increased significantly. It is worth noting that the expression regulation of the THY1 on colonocytes has an important impact on the above reactions. When THY1 on colonocytes was silenced, the DSS-induced inflammation, oxidative stress, and angiogenesis reactions were significantly reversed in co-culture system, manifested as the downregulation of inflammatory factors and oxidative stress markers, as well as the restoration of

   

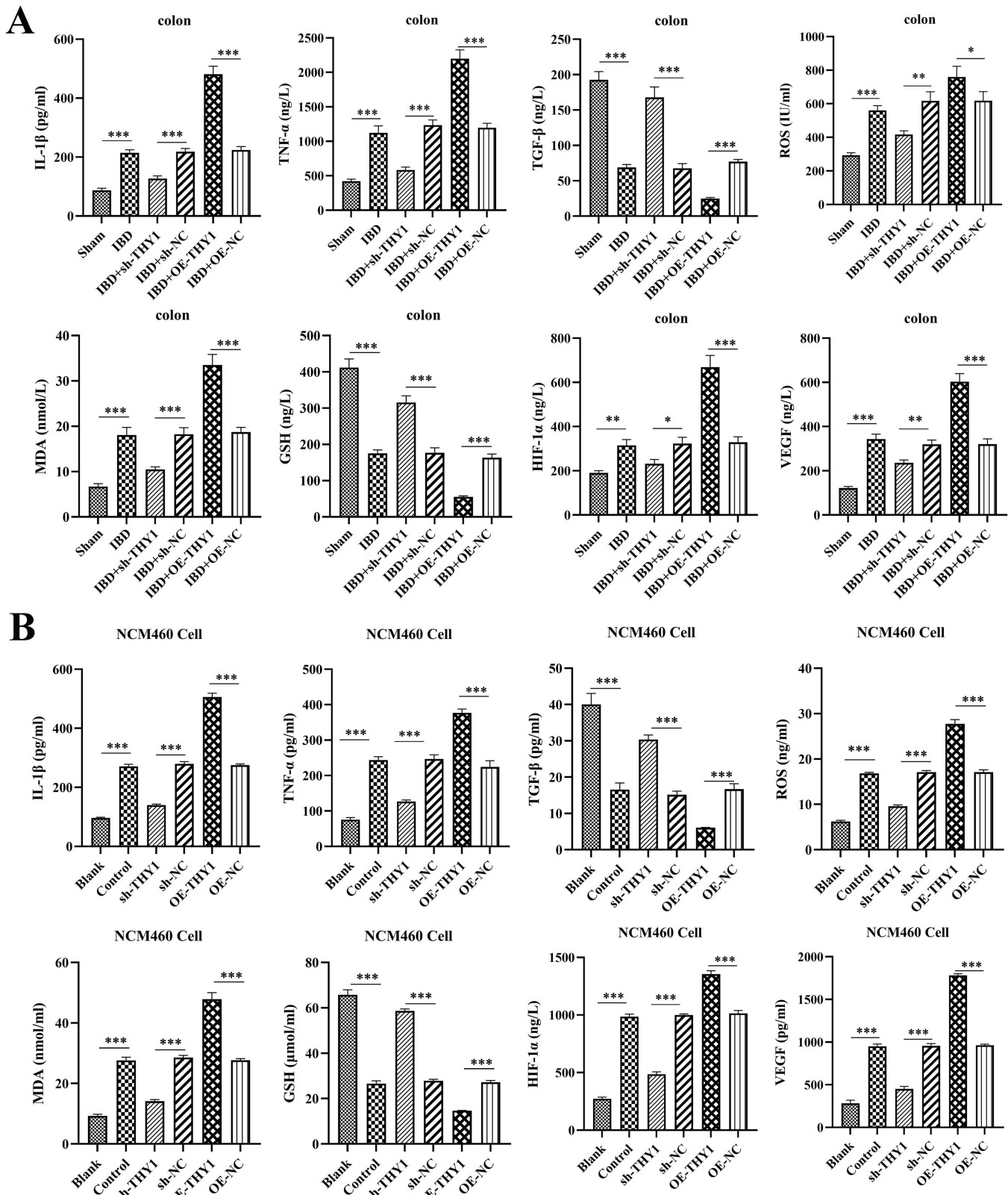

**Fig 2. THY1 promotes the pathological process of DSS-induced colitis.** (A, B) The expression levels of inflammatory factors (IL-1β, TNF-α and TGF-β), oxidative stress-related molecules (ROS, MDA and GSH) and angiogenesis-related stimulatory factors (HIF-1α and VEGF) in mouse colon

tissue samples and NCM460 cell homogenate of co-culture system were detected by enzyme-linked immunosorbent assay (ELISA). A minimum of three separate experiments were carried out and the data presented are expressed as the mean±SD. *P<0.05, **P<0.01, ***P<0.001.

anti-inflammatory and antioxidant molecules. In contrast, the overexpression of THY1 on colonocytes exacerbated these pathological reactions and further promoted the abnormal expression of factors related to inflammation, oxidative stress, and angiogenesis. These results were highly consistent with our previous findings in the in vivo colitis mouse model, further verifying the reliability of this in vitro model. To further explore the specific mechanism of THY1 in the pathological process of DSS-induced colitis, macrophage immunophenotyping were evaluated by flow cytometry. Results showed that silencing of the THY1 on colonocytes significantly induced M2 macrophage polarization while overexpression of THY1 on colonocytes significantly inhibited M2 macrophage polarization (Fig 3). Taken together, these results suggested that THY1 promotes the pathological process of DSS-induced colitis by inhibiting M2 macrophage polarization. Experimentally deplete macrophages or alter macrophage phenotypic states under THY1-OE or THY1-silenced conditions are still needed to establish such causal connection.

### THY1 promotes apoptosis in the colonic tissue of mice with DSS-induced colitis through the Bcl-2/Bax/Cleaved caspase-3 pathway

The TUNEL staining technique was used to further investigate the cell apoptosis in colon tissues, thus providing experimental evidence for revealing the key nodes regulated by THY1. TUNEL staining results showed that, compared with the control group, significant cell apoptosis occurred in the colon tissues of the DSS-treated group. Knocking down THY1 could significantly reduce the degree of tissue apoptosis. On the contrary, overexpression of THY1 further exacerbated tissue apoptosis (Fig 4A). This result indicates that THY1 plays an important role in regulating the apoptosis process of colon tissue cells. To further verify the above findings, we detected the expression levels of pro-apoptotic proteins Bax and caspase-3 and anti-apoptotic protein Bcl-2 by Western blot analysis. The results showed that knocking down THY1 could significantly down-regulate the expression of pro-apoptotic proteins Bax and cleaved caspase3, and at the same time up-regulate the expression of anti-apoptotic protein Bcl-2. In contrast, overexpression of THY1 further enhanced the expression of Bax and cleaved caspase-3 and inhibited the expression of Bcl-2, which showed a significant difference from the results of the DSS-treated group (Fig 4B).

Furthermore, the in vitro co-culture system of macrophages and intestinal epithelial cells further confirmed that compared with the Control group, interfering with the expression of THY1 could reduce the apoptosis of intestinal epithelial cells by inhibiting the expression of pro-apoptotic proteins Bax and cleaved caspase-3 and promoting the expression of anti – apoptotic protein Bcl-2. Conversely, overexpression of THY1 further exacerbated their apoptotic response of intestinal epithelial cells (Fig 5A and 5B). These data together reveal the molecular mechanism by which THY1 participates in the apoptosis of colon tissue cells by regulating the expression of apoptosis-related proteins.

### THY1 promotes angiogenesis via upregulation of HIF-1α and VEGF expression in IBD

HE staining showed that the thickness of the blood vessel wall was significantly increased in DSS-induced colitis mouse model. However, knockdown of THY1 significantly decreased the thickness of the blood vessel wall while overexpression of THY1 further increased the thickness of the blood vessel wall, indicating the regulatory effect of THY1 on the structural changes of blood vessels (Fig 6A). Quantitative real-time PCR and Western blot analysis of the mouse colon tissue samples illustrated that knockdown of THY1significantly downregulated the expression of key stimulatory factors related to angiogenesis, HIF-1α and VEGF. In contrast, overexpression of THY1 further upregulated the expression of HIF-1α and VEGF, which was in sharp contrast to the DSS group (Fig 6B and 6C).

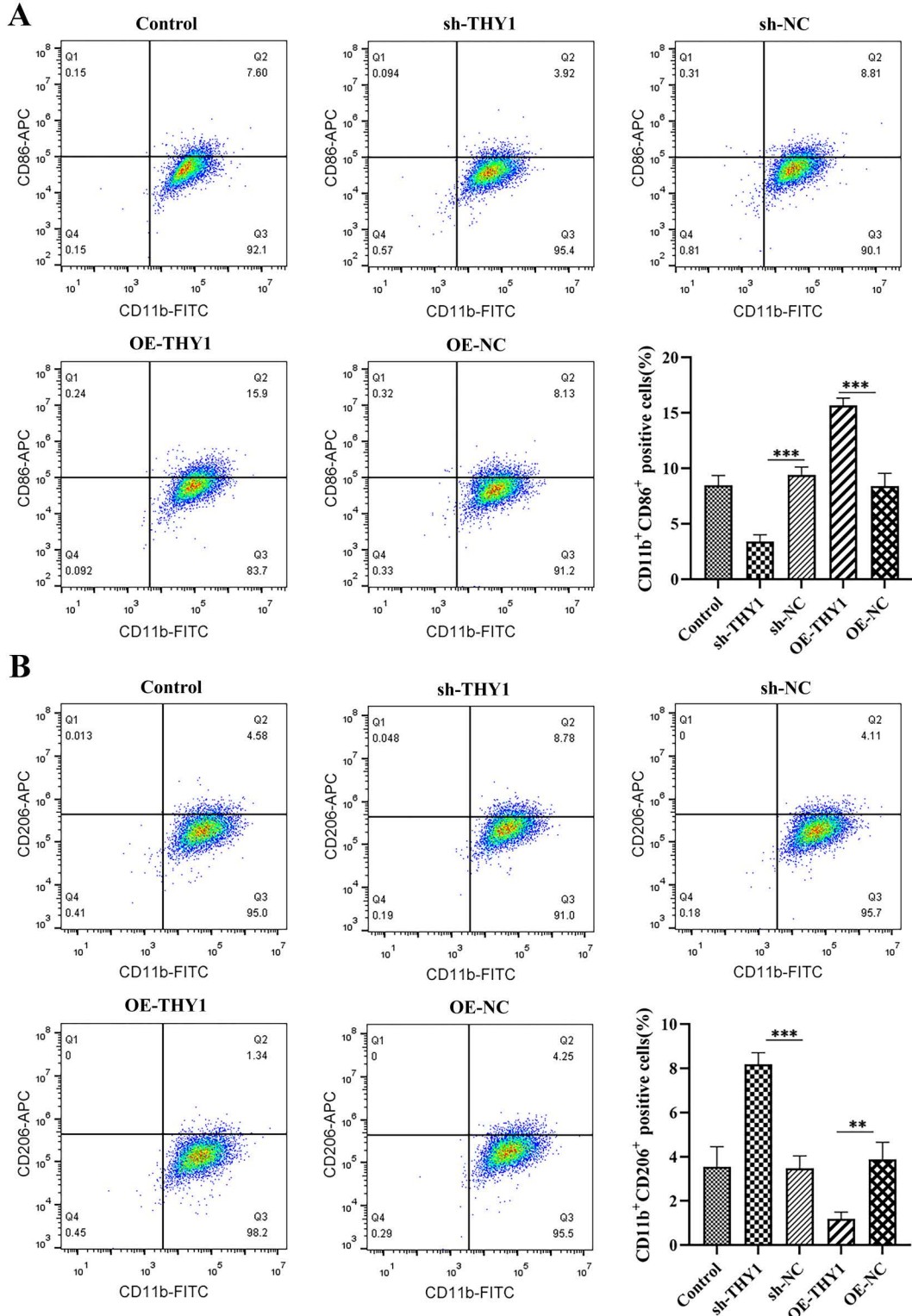

**Fig 3. THY1 inhibits M2 macrophage polarization.** (A, B) Macrophage immunophenotyping (M1 polarization and M2 polarization) were evaluated by flow cytometry in co-culture system of control group (Upper layer: THP-1 cells; Lower layer: normal NCM460 cells), sh-THY1 group (Upper layer: THP-1 cells; Lower layer: sh-THY1 treated NCM460 cells), sh-NC group (Upper layer: THP-1 cells; Lower layer: sh-NC treated NCM460 cells), OE-THY1

group (Upper layer: THP-1 cells; Lower layer: OE-THY1 treated NCM460 cells), and OE-NC treated NCM460 cells). A minimum of three separate experiments were carried out and the data presented are expressed as the mean ± SD. *P < 0.05, **P < 0.01, ***P < 0.001.

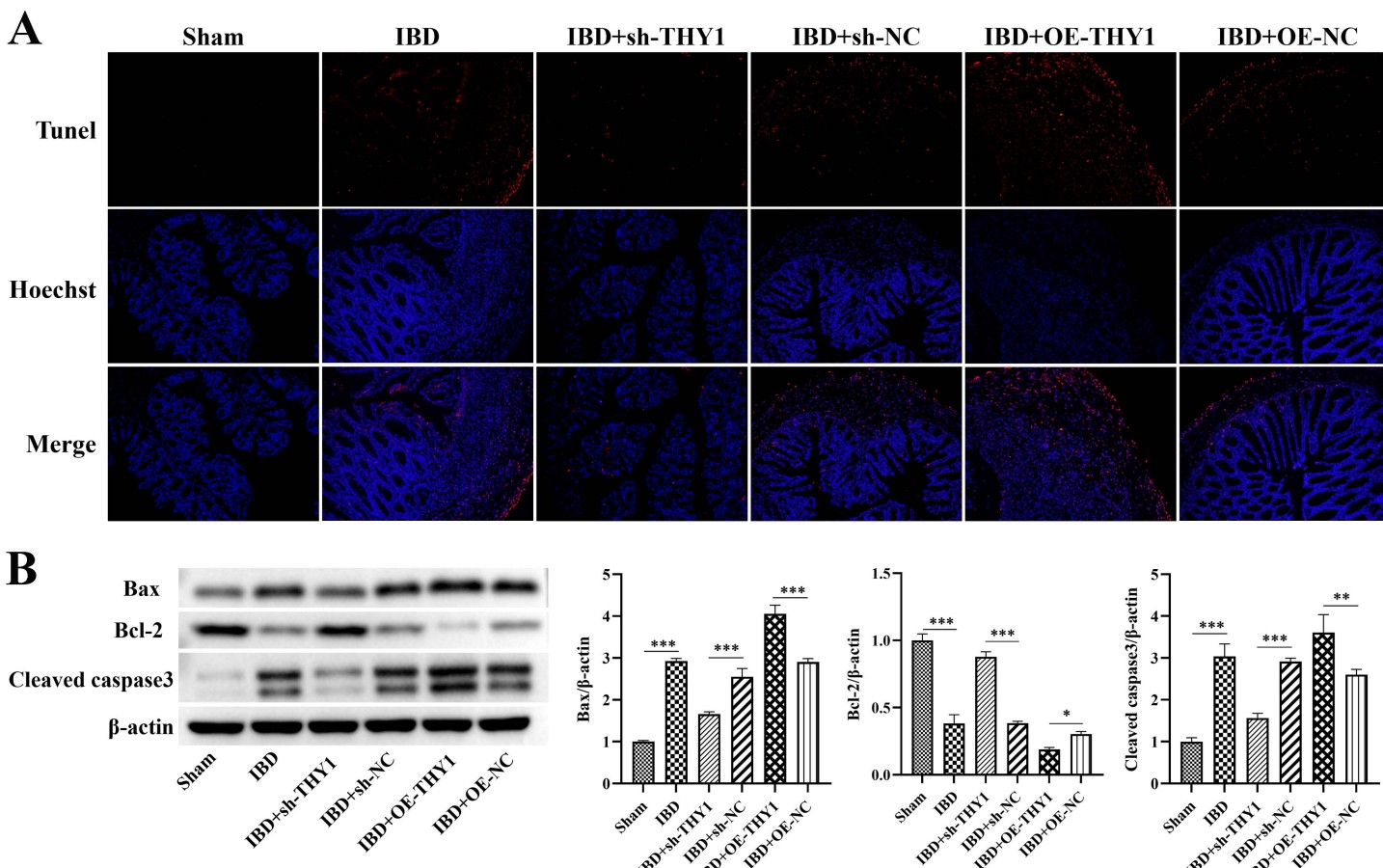

**Fig 4. THY1 promotes apoptosis in DSS-induced colitis through the Bax/cleaved caspase3/Bcl-2 pathway.** (A) Representative TUNEL staining images of DSS-induced colitis mouse in the Sham (blank control) group, the IBD model group, the IBD + OE-THY1 group, the IBD + OE-NC group. the IBD + KD-THY1 group, and the IBD + sh-NC group. (B) The expression levels of apoptosis-related proteins (Bax, Bcl-2 and cleaved caspase3) in DSS-induced colitis mouse colon tissue samples of the Sham (blank control) group, the IBD model group, the IBD + OE-THY1 group, the IBD + OE-NC group. the IBD + KD-THY1 group, and the IBD + sh-NC group. A minimum of three separate experiments were carried out and the data presented are expressed as the mean ± SD. *P < 0.05, **P < 0.01, ***P < 0.001.

To further verify the above findings, we evaluated the role of THY1 in inflammatory angiogenesis by performing a tube formation assay based on human umbilical vein endothelial cells (HUVECs) using cell culture supernatants. The experimental results showed that, compared with the blank control (culture medium) group, the supernatants of DSS-treated NCM460 cells and the co-culture supernatants of NCM460 cells and macrophages significantly promoted the tube formation of HUVECs (Fig 6D). However, when the supernatants of THY1-knockdown NCM460 cells were used, the tube formation ability of HUVECs was significantly inhibited. In contrast, the supernatants of THY1-overexpressing NCM460 cells showed a strong pro-tube formation effect, further confirming the crucial role of THY1 in regulating angiogenesis. Notably, western blot analysis of cell samples also illustrated that knockdown of THY1significantly downregulated the expression

   

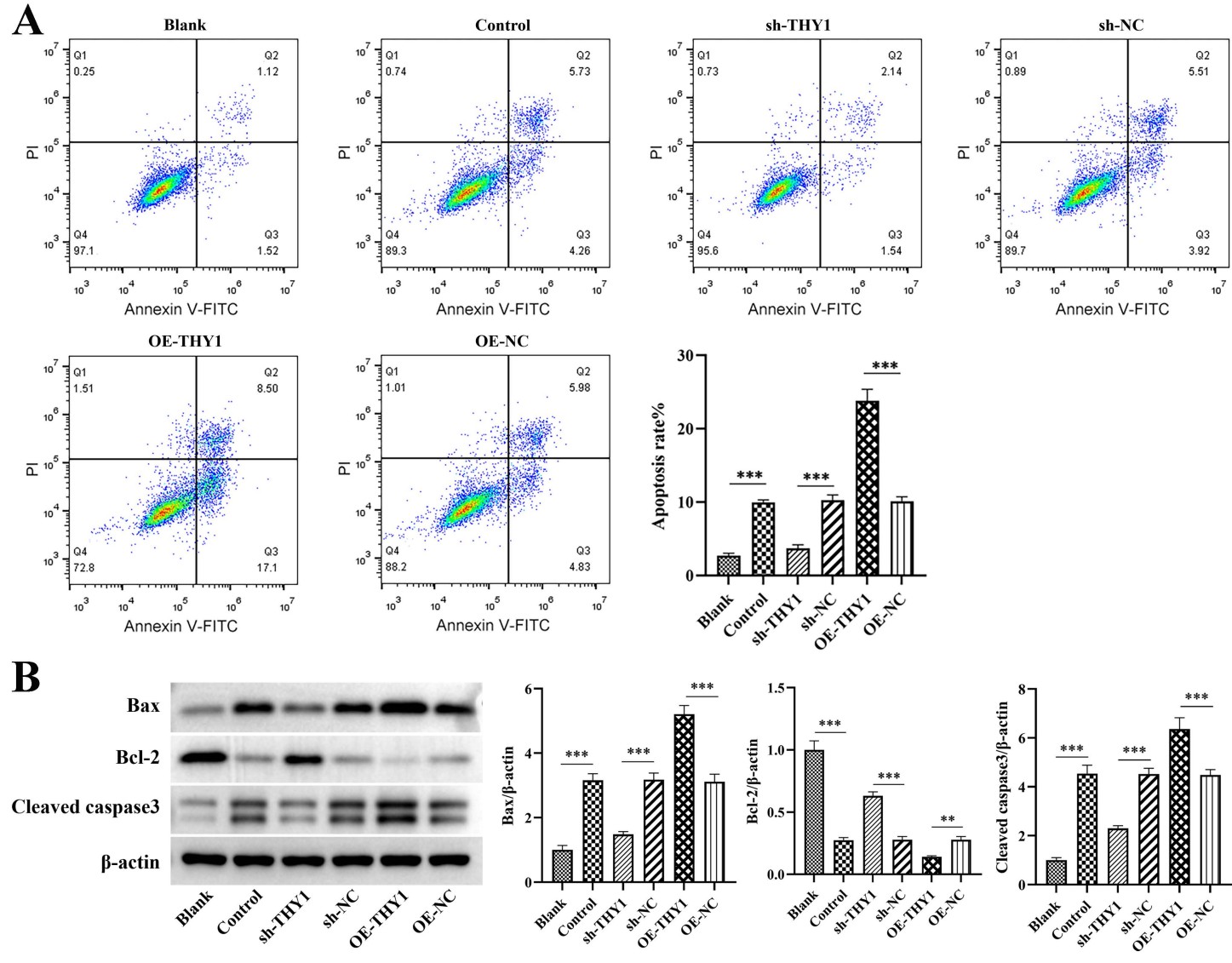

**Fig 5. THY1 promotes apoptosis in intestinal epithelial cells through the Bax/cleaved caspase3/Bcl-2 pathway.** (A) Apoptosis level and (B) apoptosis-related proteins (Bax, Bcl-2 and cleaved caspase3) expression levels of NCM460 cells in co-culture system of different treatment groups were determined by flow cytometry and western blot analysis, respectively. A minimum of three separate experiments were carried out and the data presented are expressed as the mean±SD. *P<0.05, **P<0.01, ***P<0.001.

of key stimulatory factors related to angiogenesis, HIF-1α and VEGF (Fig 6E). In contrast, overexpression of THY1 further upregulated the expression of HIF-1α and VEGF, which was in sharp contrast to the DSS group. These results together suggest that THY1 promotes angiogenesis via upregulation of HIF-1α and VEGF expression in IBD.

## Discussion

Currently, targeted therapies focusing on specific inflammatory factors and their signaling pathways, such as ustekinumab targeting IL-12 and IL-23 and small-molecule drugs tofacitinib and upadacitinib targeting the janus kinase (JAK) pathway,

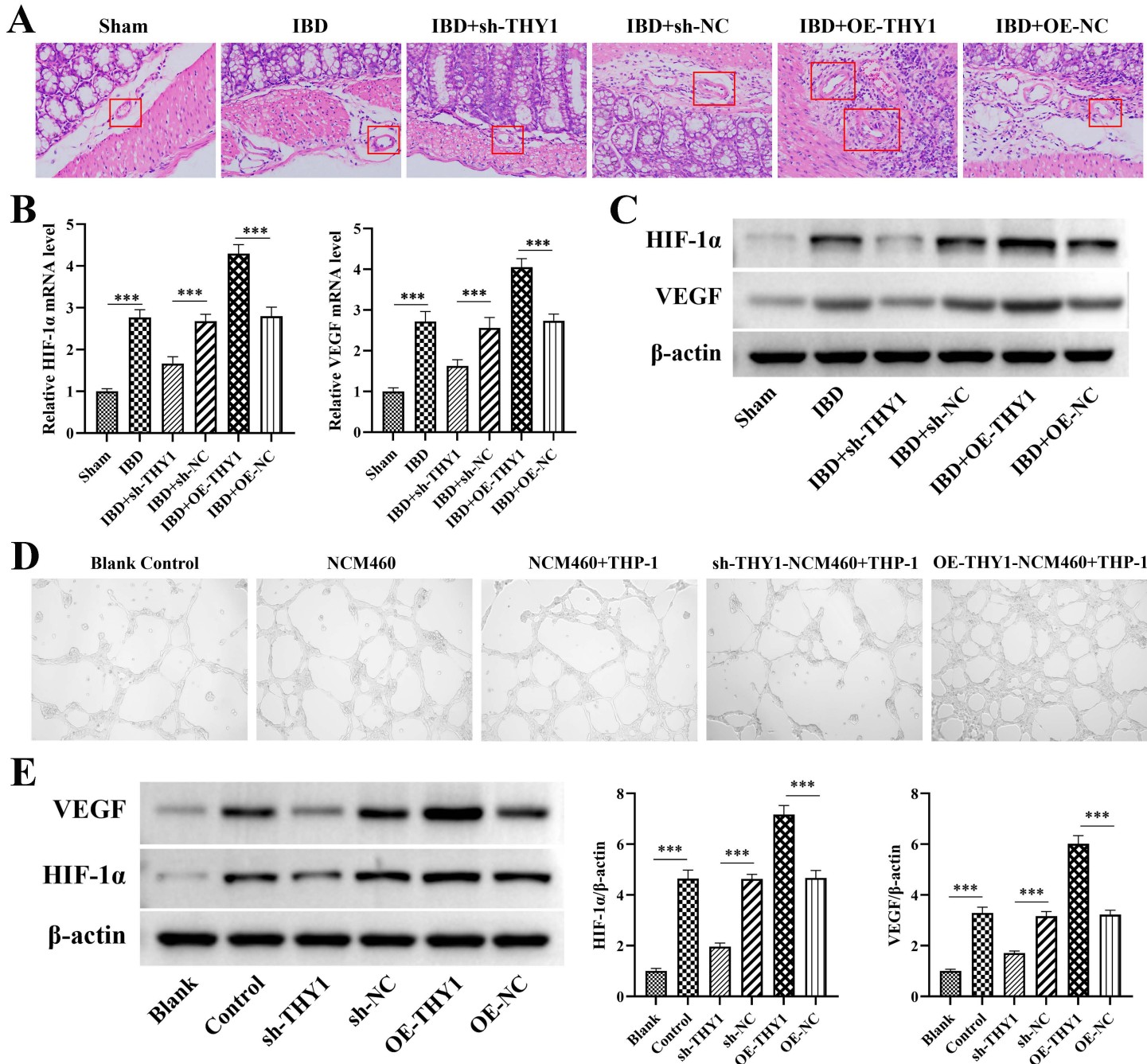

**Fig 6. THY1 promotes angiogenesis via upregulation of HIF-1α and VEGF expression in IBD.** (A) Representative H&E staining images of DSS-induced colitis mouse colon samples revealed the growth and thickness of the blood vessel wall in different treatment groups. The red square position in the image revealed the blood vessel. (B) The mRNA expression levels and (C) protein expression levels of HIF-1α and VEGF in DSS-induced colitis mouse colon tissue samples determined by quantitative real-time PCR and western blot analysis, respectively. (D) The in vitro angiogenic capacity of HUVECs stimulated with the cell culture supernatants collected from different treatment groups. (E) The protein expression levels of HIF-1α and VEGF in NCM460 cells in co-culture system of different treatment groups were determined by western blot analysis. A minimum of three separate experiments were carried out and the data presented are expressed as the mean±SD. *P<0.05, **P<0.01, ***P<0.001.

have provided new treatment options for patients with IBD [22,23]. However, although biologic targeted therapy shows significant efficacy in many patients, approximately 30% of patients do not respond to the initial treatment, and up to 50% of patients may gradually lose their treatment response over time [24]. This situation highlights the importance of further exploring potential therapeutic targets for IBD. Future research should focus on blocking the signaling pathways related to key inflammatory factors to effectively control intestinal inflammation, promote mucosal healing, and thus improve the intestinal and extra – intestinal manifestations of IBD. In this study, we revealed the crucial role of the angiogenesis-related target THY1 in the development of acute colitis induced by dextran sulfate sodium (DSS) in mice. Through in vivo and in vitro experiments, we confirmed that THY1 promotes pathological angiogenesis by regulating the HIF-1α/VEGF signaling pathway and induces apoptosis through the Bcl-2/Bax/Cleaved caspase-3 pathway, further exacerbates the pathological process of DSS-induced colitis via inhibiting M2 macrophage polarization. These findings not only deepen our understanding of the pathogenesis of IBD but also provide important theoretical basis for the development of novel treatment strategies.

Inflammatory bowel disease (IBD) is a non-specific inflammatory disease closely related to the imbalance of the body's immune system [25]. In this complex pathological process, macrophages, as key immune regulators, play a dual role: they are not only important phagocytes for phagocytosing and digesting pathogens but also play an indispensable role in maintaining the homeostasis of the intestinal microenvironment [26]. From a pathophysiological perspective, macrophages directly participate in and promote the occurrence and development of the inflammatory response by releasing a variety of inflammatory mediators (such as TNF – α, IL – 1β, etc.) [27]. However, the relationship between macrophages and IBD is not a simple linear causal relationship but presents a complex interactive model of dynamic balance. On the one hand, macrophages can exacerbate the pathological process of IBD by secreting pro-inflammatory factors; on the other hand, they can also inhibit the development of the disease by regulating the immune response [28]. For example, macrophages can release anti-inflammatory mediators (such as IL-10), thereby effectively inhibiting excessive inflammatory responses and protecting intestinal tissues from further damage [29]. This dual role indicates that macrophages are both "drivers" of inflammation and potential "guardians," and their specific functions may depend on signal regulation in the microenvironment and cell to cell interactions. Our research results show that DSS stimulation inhibits the M2 polarization of macrophages, leading to the release of pro-inflammatory factors such as TNF-α and IL-1β, which act on intestinal epithelial cells and promote cell apoptosis [30]. Notably, in vitro co-culture experiments have confirmed that interfering with the expression of THY1 promotes the M2 polarization of THP-1 cells and down-regulates the secretion of pro-inflammatory factors such as TNF-α and IL-1β, while overexpressing THY1 further inhibits the M2 polarization of THP-1 cells and promotes the secretion of pro – inflammatory factors such as TNF-α and IL-1β. All these studies indicate that THY1 plays a key role in the pathogenesis of IBD by inhibiting the polarization of M2 macrophages.

In addition, damage to the intestinal epithelial cell barrier function serves as one of the critical initiating factors in the pathogenesis of IBD. Among the various mechanisms implicated, abnormal apoptosis of intestinal epithelial cells has been identified as a pivotal driver of mucosal integrity disruption [31]. The findings of this study demonstrate that THY1 can promote intestinal epithelial cell apoptosis by modulating the Bcl-2/Bax/Cleaved caspase-3 signaling pathway. Specifically, the upregulation of Bax, a pro-apoptotic protein, coupled with the downregulation of Bcl-2, an anti-apoptotic protein, synergistically facilitates the cleavage and activation of caspase-3. This cascade ultimately exacerbates the death of intestinal epithelial cells. The intervention targeting THY1 was shown to refine the regulatory mechanisms governing intestinal epithelial cell apoptosis. This suggests that the targeted inhibition of THY1 may represent a promising therapeutic strategy to decelerate colitis progression by preserving the functional integrity of the intestinal epithelial barrier. Furthermore, the HIF-1α/VEGF axis serves as the pivotal regulatory pathway in angiogenesis. Its abnormal activation can result in increased intestinal vascular density, thereby providing nutritional support for the infiltration of inflammatory cells and exacerbating the inflammatory response [32]. This study reveals that THY1 can promote intestinal angiogenesis by upregulating the expression of HIF-1α and VEGF. These findings elucidate the upstream regulatory role of THY1 in inflammatory angiogenesis and offer a novel strategy for interrupting the "inflammation-angiogenesis" vicious cycle in IBD.

Nevertheless, this study is subject to several limitations. To begin with, the biological functions of THY1 were investigated based on mice and cell models, and its specific role in IBD still needs to be verified using human clinical samples. Secondly, the molecular interaction mechanism by which THY1 mediates angiogenesis and affects colitis remains to be fully elucidated, necessitating further investigation in future research.

## Conclusions

In conclusion, this study demonstrates that THY1 promotes the progression of DSS-induced acute colitis in mice by regulating macrophage polarization, intestinal epithelial cell apoptosis, and inflammatory angiogenesis, which provides a new perspective for the study of the pathogenesis of IBD and offers important experimental evidence for the development of clinical targeted treatment strategies. Future research can further explore the specific molecular mechanism of THY1 regulation and its application prospects in clinical translation.

### Transparent reporting on AI and AI-assisted Technologies

During the preparation of this work the authors used the AI translation function of Tianxi Personal Super Intelligent Agent (Version number: 3.5.0.10302, Lenovo Group) in order to improve the English language of the article. For example, enhance the academic tone to make the English expression clearer and more professional, and in line with the style of academic writing.

## Supporting information

**S1 Fig. Evaluation of the effectiveness of cell model establishment.** The expression levels of inflammatory factors including IL-1β, IL-6, TNF-α, and TGF-β in NCM460 cells (A) and its supernatant (B) were detected by ELISA. NCM460 cells were gradient intervention with different concentrations of DSS (e.g., 0, 0.1, 0.2, 0.5, 1, 2 µg/mL) for 12 hours. A minimum of three separate experiments were carried out and the data presented are expressed as the mean±SD. *$P<0.05$, **$P<0.01$, ***$P<0.001$.
(TIF)

**S2 Fig. THY1 promotes the pathological process of DSS-induced colitis.** The expression levels of inflammatory factors (IL-1β, TNF-α and TGF-β), oxidative stress-related molecules (ROS, MDA and GSH) and angiogenesis-related stimulatory factors (HIF-1α and VEGF) in mouse blood serum samples of different treatment groups were detected by enzyme-linked immunosorbent assay (ELISA). A minimum of three separate experiments were carried out and the data presented are expressed as the mean±SD. *$P<0.05$, **$P<0.01$, ***$P<0.001$.
(TIF)

**S3 Fig. The full uncropped Gels and Blots images used in this study.**
(PDF)

**S4 Fig. Original data sets for Figures in this study.**
(ZIP)

## Acknowledgments

No additional acknowledgements beyond the listed authors are applicable to this study.

## Author contributions

**Conceptualization:** Pengliang Zhang, Xianmin Liu.

**Data curation:** Pengliang Zhang.

**Formal analysis:** Pengliang Zhang.

**Funding acquisition:** Yingjian Zhang.

**Investigation:** Xianmin Liu, Shuang Chen.

**Methodology:** Pengliang Zhang.

**Project administration:** Yingjian Zhang.

**Resources:** Xianmin Liu, Shuang Chen.

**Software:** Pengliang Zhang.

**Supervision:** Yingjian Zhang.

**Validation:** Xianmin Liu, Shuang Chen.

**Visualization:** Pengliang Zhang.

**Writing – original draft:** Pengliang Zhang.

**Writing – review & editing:** Xianmin Liu, Shuang Chen, Yingjian Zhang.

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
