## [Decision Letter · Decision Letter 0]

10 Mar 2026

PONE-D-26-02176The role and targeting potential analysis of angiogenesis-related target THY1 in DSS-induced acute colitis in micePLOS One

Dear Dr. Zhang,

Thank you for submitting your manuscript to PLOS ONE. After careful consideration, we feel that it has merit but does not fully meet PLOS ONE’s publication criteria as it currently stands. Therefore, we invite you to submit a revised version of the manuscript that addresses the points raised during the review process.

We look forward to receiving your revised manuscript.

Kind regards,

Subhasis Barik

Academic Editor

PLOS One

Journal Requirements:

Additional Editor Comments:

Both the reviewers raised multiple issues that need to be addressed in the revised manuscript.

Reviewers' comments:

Reviewer's Responses to Questions

**Comments to the Author**

1. Is the manuscript technically sound, and do the data support the conclusions?

Reviewer #1: Partly

Reviewer #2: Yes

2. Has the statistical analysis been performed appropriately and rigorously? 

Reviewer #1: Yes

Reviewer #2: Yes

3. Have the authors made all data underlying the findings in their manuscript fully available?

Reviewer #1: Yes

Reviewer #2: No

4. Is the manuscript presented in an intelligible fashion and written in standard English?

Reviewer #1: Yes

Reviewer #2: No

5. Review Comments to the Author

Reviewer #1: The manuscript entitled “The role and targeting potential analysis of angiogenesis-related target THY1 in DSS-

induced acute colitis in mice” explores the role of THY1 in two aspects of DSS-induced colitis: alterations in colonocyte biology and changes in macrophage phenotypes. While the study apparently focuses on how THY1 regulates the phenotypic alterations of these cells that lead to colitis pathology, there are specific points which are not clear to me in terms of logic, as outlined below.

Major points

1- How well-established is the cell line-based model of IBD (lines 159-161)? Authors need to cite references.

2- How could the authors guarantee that the lentiviral vector-mediated delivery of the constructs could solely be targeted to colonocytes and not any stromal or immune cells around them? This is particularly important as THY1 expression on stromal cells has a crucial role in colitis pathogenesis (reference: https://doi.org/10.4049/jimmunol.212.supp.1326.5032).

3- The 2nd point of the results (lines 303-348) only shows that THY1 causes definite changes in macrophage phenotypes parallel to the aggravation of DSS-induced colitis in mice. How could the authors draw a causal inference in the statement “THY1 promotes the pathological process of DSS-induced colitis through inhibiting M2 macrophage polarization?” To establish such causal connection, authors need to experimentally deplete macrophages or alter macrophage phenotypic states under THY1-OE or THY1-silenced conditions, and repeat the same experiments in vivo.

4- In the same point, the authors seemingly check several different aspects (cytokines, ROS, angiogenesis markers) related to macrophages. However, there is no definitive proof that the changes in these markers due to THY1 manipulations are solely, or at least majorly, related to their altered output from macrophages in vivo. In addition, why they check so many aspects together without any apparently clear hypothesis is not understandable.

5- Again, in the coming sections, authors assess colonocyte apoptosis and angiogenesis marker levels without any clear connection to the previous sections. The entire story looks like an ensemble of experiments without any proper logical link among them.

6- Although the authors check multiple phenotypic effects of THY1 manipulations on colonocytes, how each of them affects colitis has not been checked. So the entire set of findings by large stays correlative.

7- Why do the authors check ROS-related phenotypes (ROS, MDA, GSH etc) in cell supernatants? It would have been more physiologically relevant if they had done the experiment with cell homogenates/lysates.

8- None of the experiments involve DSS-conditioning of the THP-1 cells, and only involves co-culturing them with colonocytes of different phenotypes. This violates the physiology of colitis, where macrophages also encounter DSS, and are likely affected by it to some extent as well (reference: https://doi.org/10.1691/ph.2016.6688).

Minor points

1- It would have been preferable if the authors would have shown the time kinetics of THY1 expression in colon along the course of DSS treatment in mice and afterwards.

2- For apoptosis assay, how could the authors resuspend the cells in 1 microliter binding buffer? I suspect it is a typing error, and needs to be taken care of. Furthermore, didn’t they use any strategy for gating live/dead cells differentially in their flow cytometry data for macrophages?

3- If colonocytes were the target, why did the authors opt for lentivirus injection intravenously?

4- Why didn't the authors opt for showing the intestines from mice groups photographically? It might have been more convincing than putting up a graph.

5- A much better way to represent the data in fig. 3 would have been to gate on the CD11b+ cells, followed by the assessment of CD86 and CD206 simultaneously in them.

6- For cleaved caspase-3 immunoblots, total caspase-3 level needs to be demonstrated as well.

7- Was VEGF levels quantified with or without Golgi-stop/Golgi-plug treatment?

8- For mice experiments, plotting individual data points is always preferable. In addition, it is better to plot SEM rather than SD for showing data corresponding to biological replicates.

9- Did the authors perform normality tests before doing parametric statistical tests?

10- In the cases where authors have claimed “THY1 does so..” and on, should be reframed as “colonocyte-expressed THY1 does so…” or “THY1 on colonocytes does so….”. This will specify the findings.

Reviewer #2: The study by Pengliang Zhang et al claim to identify a novel role of THY 1 in DSS induced colitis. They found out that THY1 promotes inflammatory angiogenesis that exacerbate the disease. The methods and models employed enrich the study and the data gathered collectively reinforces the key hypothesis of the study. The hard work is commendable. However some clarifications are needed.

1.) The study doesn’t appear to be unique and the first of it’s kind as evident from the sentence “In this study, we first revealed the crucial role of the angiogenesis-related target THY1 in the development of acute colitis”. One similar study “The Journal of Immunology, Volume 212, Issue 1,1326_5032, https://doi.org/10.4049/jimmunol.212.supp.1326.5032” published in 2024 had a similar objective and conclusion. Many more can be cited. Hence, it appears that the authors reinforces and substantiate the earlier studies.

2.) Use of a variety of techniques and models often overwhelm the readers. The design could have been much simplified and succinctly presented to make reading and assimilation easier.

3.) Although the authors state the use of AI for improving English, the English still need ample revision.

4.) In the co-culture experiment, choice of human cell lines NCM460 and THP-1 is not accurate. A co-culture of Mice primary intentional epithelial cells and peritoneal macrophages would have been more appropriate to establish and validate the in vitro model

5.) What is the natural ligand for THY1 and are there any inhibitors of THY1 that may have been used alongside or instead of sh-THY1?

The study presents a detailed evaluation of the effect of THY1 in colitis, which needs substantial refinement for enhanced clarity in light of the points raised.

6. PLOS authors have the option to publish the peer review history of their article (what does this mean?). If published, this will include your full peer review and any attached files.

Reviewer #1: **Yes:** Soumyadeep Mukherjee

Reviewer #2: No

---

## [Author Response · Author response to Decision Letter 1]

1 Apr 2026

Dear editor,

Thank you very much for your letter enclosing the academic editor and reviewers’ comments. We appreciate the academic editor and reviewers’ comments for further improving our manuscript PONE-D-26-02176. Those comments are all valuable and very helpful for revising and improving our paper. We have carefully reviewed the comments and have revised the manuscript accordingly, which we hope meet with approval. In the following pages are our point-by-point responses or a rebuttal against each point raised by the academic editor and the reviewers.

We shall look forward to hearing from you at your earliest convenience.

With regards,

Yingjian Zhang

Responses to the academic editor and reviewers’ comments:

Journal Requirements:

Response: We have carefully revised our manuscript body formatting according to the PLOS ONE style templates. Manuscript Organization, heading style and File Naming for Figures have been proofread to meets PLOS ONE's style requirements.

Response: We have deleted the "Ethical Considerations" section from our manuscript according to the Journal Requirement.

Response: The full uncropped Gels and Blots images used in this study have been provided as S3 Fig in Supporting Information.

Response: Checked.

Response: We have included captions for our Supporting Information files at the end of the manuscript, and updated any in-text citations to match accordingly.

Response: Checked.

Reviewer #1: The manuscript entitled “The role and targeting potential analysis of angiogenesis-related target THY1 in DSS-

induced acute colitis in mice” explores the role of THY1 in two aspects of DSS-induced colitis: alterations in colonocyte biology and changes in macrophage phenotypes. While the study apparently focuses on how THY1 regulates the phenotypic alterations of these cells that lead to colitis pathology, there are specific points which are not clear to me in terms of logic, as outlined below.

Major points

1- How well-established is the cell line-based model of IBD (lines 159-161)? Authors need to cite references.

Response: We are very appreciating the reviewer’s comments. According to the reviewer’s suggestion, we have added the relevant reference (https://doi.org/10.5114/ceji.2025.149579) to clarify the maturity of the colitis cell model in our study. According to cited reference, colitis cell models were constructed by treating human normal colonic epithelial cells (FHC) with different concentrations of lipopolysaccharide (LPS) (1, 3, 5, 10, and 15 ng/ml). In our study, normal human intestinal epithelial cell line NCM460 were gradient intervention with different concentrations of dextran sulfate sodium (DSS) (e.g., 0, 0.1, 0.2, 0.5, 1, 2 μg/mL) for 12 hours to establish an IBD cell model. And the expression levels of inflammatory factors including IL-1β, IL-6, TNF-α, and TGF-β in cells were determined to confirm the successful establishment of the model. Our results indicate that an inflammatory bowel disease (IBD) cell model was successfully established based on this cell line.

2- How could the authors guarantee that the lentiviral vector-mediated delivery of the constructs could solely be targeted to colonocytes and not any stromal or immune cells around them? This is particularly important as THY1 expression on stromal cells has a crucial role in colitis pathogenesis (reference: https://doi.org/10.4049/jimmunol.212.supp.1326.5032).

Response: We are very appreciating the reviewer’s comments. We cannot guarantee that the lentiviral vector-mediated delivery of the constructs only targets colonic cells without targeting any surrounding stromal cells or immune cells. However, our study aims to evaluate the role of THY1 in the pathogenesis of DSS-induced colitis in mice, rather than the role of THY1 expression in a single cell type. Therefore, we used lentiviral vector-mediated construct delivery to regulate the expression of THY1 in the colonic tissue of mice, as shown in the results of our Fig 1B.

3- The 2nd point of the results (lines 303-348) only shows that THY1 causes definite changes in macrophage phenotypes parallel to the aggravation of DSS-induced colitis in mice. How could the authors draw a causal inference in the statement “THY1 promotes the pathological process of DSS-induced colitis through inhibiting M2 macrophage polarization?” To establish such causal connection, authors need to experimentally deplete macrophages or alter macrophage phenotypic states under THY1-OE or THY1-silenced conditions, and repeat the same experiments in vivo.

Response: We are very appreciating the reviewer’s comments. We acknowledge that it is necessary to add the experiments suggested by the reviewer to draw the causal inference that "THY1 promotes the pathological process of DSS-induced colitis through inhibiting M2 macrophage polarization." Therefore, we have changed the second point of the results to "THY1 promotes the pathological process of DSS-induced colitis and inhibits M2 macrophage polarization."

4- In the same point, the authors seemingly check several different aspects (cytokines, ROS, angiogenesis markers) related to macrophages. However, there is no definitive proof that the changes in these markers due to THY1 manipulations are solely, or at least majorly, related to their altered output from macrophages in vivo. In addition, why they check so many aspects together without any apparently clear hypothesis is not understandable.

Response: We are very appreciating the reviewer’s comments. The detection of cytokines, reactive oxygen species, and angiogenesis markers is to clarify the effect of THY1 on the pathological process of DSS-induced colitis. First, we found the regulatory effect of THY1 on cytokines, reactive oxygen species, and angiogenesis markers through in vivo experiments. Given that macrophages play an important immunomodulatory role in the pathological process of colitis, in this study, we further elucidated the molecular mechanism from the perspective of macrophage immunomodulation through an in vitro co-culture system of intestinal epithelial cells and macrophages.

5- Again, in the coming sections, authors assess colonocyte apoptosis and angiogenesis marker levels without any clear connection to the previous sections. The entire story looks like an ensemble of experiments without any proper logical link among them.

Response: We are very appreciating the reviewer’s comments. This study aimed to explored and demonstrated the potential role of THY1 in the pathological mechanism of IBD from multiple perspectives based on DSS-induced acute colitis in mice and colonocytes models. Existing studies have shown that the impairment of the intestinal epithelial cell barrier function caused by colonic cell apoptosis is a key initiating factor in the pathogenesis of IBD, and abnormal angiogenesis can provide nutritional support for the infiltration of inflammatory cells and exacerbate the inflammatory response. Therefore, on the basis of revealing the potential role of THY1 in the pathological mechanism of IBD from the perspective of macrophage immunomodulation, we further revealed the role of THY1 in the pathogenesis of IBD from the perspectives of colonocyte apoptosis and abnormal angiogenesis, aiming to provide theoretical support for the development of THY1-targeted IBD treatment strategies. We acknowledge that the proper logical link among them was not shown in this study. However, this study has completely revealed the role and targeting potential of angiogenesis-related target THY1 in DSS-induced acute colitis in mice. Moreover, we have pointed out the limitations of this study in the discussion section.

6- Although the authors check multiple phenotypic effects of THY1 manipulations on colonocytes, how each of them affects colitis has not been checked. So the entire set of findings by large stays correlative.

Response: We are very appreciating the reviewer’s comments. Regarding the limitations of this study pointed out by the reviewer, it should be noted that in this study, we aimed to explored and demonstrated the potential role of THY1 in the pathological mechanism of IBD from multiple perspectives based on DSS-induced acute colitis in mice and colonocytes models, providing theoretical support for the development of THY1-targeted IBD treatment strategies. In addition, in the discussion section, we also pointed out the limitations of this study, including that the molecular mechanism by which THY1 mediates angiogenesis and affects colitis needs to be fully elucidated.

7- Why do the authors check ROS-related phenotypes (ROS, MDA, GSH etc) in cell supernatants? It would have been more physiologically relevant if they had done the experiment with cell homogenates/lysates.

Response: We are very appreciating the reviewer’s comments. We already have done the experiment with cell homogenates/lysates, which was shown as S3 in Supporting Information files. According to the review’s comments, we have replaced the detection results in the cell supernatant with those in the cell homogenate/lysate, as revised in Fig 2B.

8- None of the experiments involve DSS-conditioning of the THP-1 cells, and only involves co-culturing them with colonocytes of different phenotypes. This violates the physiology of colitis, where macrophages also encounter DSS, and are likely affected by it to some extent as well (reference: ).

Response: We are very appreciating the reviewer’s comments. In our co-culture system of intestinal epithelial cells and macrophages, we aimed to investigate the immunomodulatory role of macrophages in the pathological process of colitis regulated by THY1. Therefore, the DSS-conditioning of the THP-1 cells group was not necessary.

Minor points

1- It would have been preferable if the authors would have shown the time kinetics of THY1 expression in colon along the course of DSS treatment in mice and afterwards.

Response: We are very appreciating the reviewer’s comments. In this study, we used lentiviral vector-mediated construct delivery to regulate the expression of THY1 in the colonic tissue of mice. Through end-point mode evaluation, we further explored the potential role of THY1 in the pathological mechanism of DSS-induced colitis. It is worth noting that the temporal dynamics of THY1 expression in colonic tissues were not the core presentation content of this study, so they were not presented in detail.

2- For apoptosis assay, how could the authors resuspend the cells in 1 microliter binding buffer? I suspect it is a typing error, and needs to be taken care of. Furthermore, didn’t they use any strategy for gating live/dead cells differentially in their flow cytometry data for macrophages?

Response: We are very appreciating the reviewer’s comments. By examining the data from our apoptosis assay, we found that the cells were resuspended in 300 microliters of binding buffer. Therefore, we have revised the error in text as follows: “Cells were harvested, rinsed with PBS, and resuspended in 300 microliters of Annexin V binding buffer to achieve a single-cell suspension.” For macrophage immunophenotyping analysis, we adopted a gating strategy to select live CD11b+ macrophages. Specifically, we first excluded debris and doublet cells using forward scatter (FSC)/side scatter (SSC) parameters, then selected live CD11b+ monocytes/macrophages. We have added the following modification to the text: “The gating strategy adopted a step-by-step approach: First, use the forward scatter (FSC)/side scatter (SSC) parameters to exclude debris and doublet cells, and then select live CD11b+ monocytes/macrophages.”

3- If colonocytes were the target, why did the authors opt for lentivirus injection intravenously?

Response: We are very appreciating the reviewer’s comments. Our study aims to evaluate the role of THY1 in the pathogenesis of DSS-induced colitis in mice, rather than the role of THY1 expression in a single cell type. Therefore, we used lentiviral vector-mediated construct delivery to regulate the expression of THY1 in the colonic tissue of mice, as shown in the results of our Fig 1B.

4- Why didn't the authors opt for showing the intestines from mice groups photographically? It might have been more convincing than putting up a graph.

Response: We are very appreciating the reviewer’s comments. Since all groups of colons were not arranged and photographed during the mouse dissection, we did not visually present pictures of intestinal tissues. However, we have taken comparative photos of the THY1 overexpression group and the Sham group, as shown below. We guarantee that all data is real and reliable, and no photo display is required.

5- A much better way to represent the data in fig. 3 would have been to gate on the CD11b+ cells, followed by the assessment of CD86 and CD206 simultaneously in them.

Response: We are very appreciating the reviewer’s comments. In this study, we have adopted a gating strategy to select live CD11b+ macrophages for macrophage immunophenotyping analysis. According to the review’s second minor point, we have added the following modification to the method section: “The gating strategy adopted a step-by-step approach: First, use the forward scatter (FSC)/side scatter (SSC) parameters to exclude debris and doublet cells, and then select live CD11b+ monocytes/macrophages.” In addition, given the large number of groups included in this study, incorporating the gating data into the current Fig 3 would result in an excessive number of panels within the overall data graph. This would compromise the clarity and effectiveness of the presentation. Based on this, we believe that there is no need to add additional gating data at present to ensure the simplicity of the chart and the effectiveness of information transmission.

6- For cleaved cas

---

## [Decision Letter · Decision Letter 1]

13 May 2026

The role and targeting potential analysis of angiogenesis-related target THY1 in DSS-induced acute colitis in mice

PONE-D-26-02176R1

Dear Dr. Zhang,

We’re pleased to inform you that your manuscript has been judged scientifically suitable for publication and will be formally accepted for publication once it meets all outstanding technical requirements.

Kind regards,

Subhasis Barik

Academic Editor

PLOS One

Additional Editor Comments (optional):

Reviewers' comments:

Reviewer's Responses to Questions

**Comments to the Author**

1. If the authors have adequately addressed your comments raised in a previous round of review and you feel that this manuscript is now acceptable for publication, you may indicate that here to bypass the “Comments to the Author” section, enter your conflict of interest statement in the “Confidential to Editor” section, and submit your "Accept" recommendation.

Reviewer #1: All comments have been addressed

Reviewer #2: All comments have been addressed

2. Is the manuscript technically sound, and do the data support the conclusions?

Reviewer #1: Yes

Reviewer #2: Yes

3. Has the statistical analysis been performed appropriately and rigorously? 

Reviewer #1: Yes

Reviewer #2: I Don't Know

4. Have the authors made all data underlying the findings in their manuscript fully available?

Reviewer #1: Yes

Reviewer #2: Yes

5. Is the manuscript presented in an intelligible fashion and written in standard English?

Reviewer #1: Yes

Reviewer #2: Yes

6. Review Comments to the Author

Reviewer #1: All comments from the previous round have been addressed. I recommend the manuscript for acceptance in its current form.

Reviewer #2: The revised manuscript is fair enough for acceptance. The authors have responded to the reviewer comments with clarity.

7. PLOS authors have the option to publish the peer review history of their article (what does this mean?). If published, this will include your full peer review and any attached files.

Reviewer #1: No

Reviewer #2: No

---

## [Editor Report · Acceptance letter]

PONE-D-26-02176R1

PLOS One

Dear Dr. Zhang,

I'm pleased to inform you that your manuscript has been deemed suitable for publication in PLOS One. Congratulations! Your manuscript is now being handed over to our production team.

Kind regards,

on behalf of

Dr. Subhasis Barik

Academic Editor

PLOS One